# An Integrative Phylogenetic Analysis of the Genus *Rhynchium* Spinola (Hymenoptera: Vespidae: Eumeninae) from China Based on Morphology, Genomic Data and Geographical Distribution

**DOI:** 10.3390/insects16020217

**Published:** 2025-02-16

**Authors:** Yun-Lian Peng, Shu-Lin He, Bin Chen, Ting-Jing Li

**Affiliations:** Chongqing Key Laboratory of Vector Control and Utilization, Institute of Entomology and Molecular Biology, College of Life Science, Chongqing Normal University, Chongqing 401331, China; 2022110513055@stu.cqnu.edu.cn (Y.-L.P.); shulinhe@cqnu.edu.cn (S.-L.H.); bin.chen@cqnu.edu.cn (B.C.)

**Keywords:** *Rhynchium*, morphological identification, molecular species delimitation, color patterns, geographical distribution

## Abstract

By combining morphology with population genetic differentiation, phylogenetic relationship and geographical distribution, the taxonomic confusion of the genus *Rhynchium* Spinola in China was resolved. Three Chinese species, namely *Rhynchium carnaticum*, *Rhynchium quinquecinctum*, and *Rhynchium brunneum,* were verified, of which *R. carnaticum* is newly recorded and *R. brunneum* is widely distributed in China. Their morphologic features for classification are updated. The color patterns of the widely distributed species *R. brunneum* are diverse and were preliminarily analyzed in this study, which provides a useful reference for future explorations of the evolution of the family Vespidae.

## 1. Introduction

The genus *Rhynchium* (Spinola, 1806) of the subfamily Eumeninae, generally with a stout and big-sized body, plays an important role in controlling pests in agroforestry ecosystems [1,2]. To date, 21 species have been recorded and are relatively widespread in the Old World, particularly in the Middle East and Africa. Some widely distributed taxa, such as *R. haemorrhoidale* (Fabricius, 1775), *R. marginellum* (Fabricius, 1793), and *R. quinquecinctum* (Fabricius, 1787), exhibit diverse color morphs and subspecific variations [3,4,5]. In China, two species, *R*. *brunneum* (Fabricius, 1793) and *R*. *quinquecinctum* (Fabricius, 1787), were reported from several provinces in previous studies [6,7,8,9]. Over the past two decades, a number of *Rhynchium* specimens have been accumulated across China. These specimens exhibit marked color variations, some of which display transitional coloration. The transitional coloration complicates interspecific differentiation as noted by Giordani Soika (1986) [10]: “la descrizione del *quinquecinctum* corrisponde bene a molti esemplari della China, mentre gli esemplari a me noti dell’ India, Burma e Tonkino hanno la colorazione del *brunneum*”. In the genus *Rhynchium,* colors and punctures are usually key differentiating features, especially for *R*. *brunneum* (Fabricius, 1793) and *R*. *quinquecinctum* (Fabricius, 1787) [3,7,10,11]. However, color and density of punctures often vary within the species in the family Vespidae, which adds confusion to the taxonomic identification. For instance, *Vespa velutina* (Lepeletier, 1836) has two color patterns, and *Vespa analis* (Fabricius, 1775) has three color patterns [12]. Moreover, the species *Polistes* (*Polistella*) *strigosus* (Bequaert, 1940) and *Euodynerus dantici* (Rossi, 1790) also exhibit numerous color patterns [13,14]. To address the complex differentiating challenges of the genus *Rhynchium* in the process of practical application, integration of morphological differences containing genitalia, population genetic differentiation, phylogenetic relationship and geographical distributions will be necessary.

Molecular data are scarce in the genus *Rhynchium*. The cytochrome oxidase subunit I (mtDNA COI) gene sequence in the subfamily Eumeninae is relatively conserved and suitable for molecular barcoding [15], which can be used for interspecific molecular identification. In addition, increasingly used markers such as universal single-copy orthologs (USCOs) were obtained for phylogenetic analysis when *COI* sequences could not resolve the disputation relationships [16]. USCOs selected from OrthoDB orthologous groups that contain genes present as single-copy orthologs in at least 90% of the species have provided a solid basis for inferring species phylogenetics [17]. Although the mitochondrial genome reflects evolutionary relationships by maternal inheritance and has unique advantages in studying the recent evolution and maternal genetic structure of species [18,19,20], the mitochondrial genome may give misleading results because of the limited genetic information in mitochondria [21]. In contrast, USCOs reflect the evolutionary relationships of species at the level of the nuclear genome, including information about biparental inheritance and for studying complex evolutionary events, such as whole-genome duplication, gene loss and horizontal gene transfer [22,23]. In order to accurately solve the existing taxonomic problems of *Rhynchium* in China, both mitochondrial genomes and USCOs will be applied in this study.

## 2. Materials and Methods

### 2.1. Morphological Analysis

Specimens were deposited in the Institute of Entomology and Molecular Biology, Chongqing Normal University (CNU) and the Natural History Museum of UK (NHM). A total of more than 600 dried specimens and alcohol-preserved specimens were examined for morphological analysis. The morphological and color characteristics were examined and the male genitalia were dissected under an Olympus SZ61 stereomicroscope. Photos were taken with a Keyence VHX-5000 digital camera. The plates were compiled with Photoshop 2020.

Through preliminary morphological classification with the exclusion of color and geographic differences, most of our specimens could be roughly divided into three groups by specific distinguishing features in eumenids: (A) Inner side of posterior ocellus without bulge, and posterior ocellus completely visible in frontal view; (B) Cephalic foveae in female wider than the distance between posterior ocelli; mesoscutum posteriorly and scutellum dull, densely punctate; metanotum hardly compressed medially; (C) Cephalic foveae in female as wide as the distance between posterior ocelli, mesoscutum posteriorly and scutellum anteriorly impunctate, more or less polished and metanotum compressed medially. However, some specimens were in a transitional state of both color and punctures between (B) and (C). In order to further distinguish these species, we dissected the genitalia of male specimens including the above categories and other intermediate states, except (A), which lacked male specimens.

### 2.2. Sampling and Sequencing

The 35 newly sequenced samples, preserved in 95% ethanol and stored at −20 °C, were sent to Novogene Bioinformatics Technology Co., Ltd. (Tianjin, China) for DNA extraction. Sequencing was carried out on an Illumina platform with PE150, generating approximately 6 Gb of raw data for each sample. Subsequently, the raw data were processed by fastp v0.23.2 [24] with specific parameters: “-g -q 5 -u 50 -n 15 -l 150 --overlap_diff_limit 1 --overlap_diff_percent_limit 10”, to ensure data quality and obtain clean data suitable for subsequent analyses. The whole genome sequences of two samples were downloaded from NCBI (NCBI accession numbers: SAMN36845277 and SAMN36845333). A total of 37 samples of whole-genome sequencing data were obtained. The geographical distribution and NCBI accession numbers of the 37 samples are detailed in Figure 1 and Appendix A, respectively.

### 2.3. Mitogenome Assembly and Dataset Generation

Mitogenome assembly was performed using MitoZ v3.4 [25] from the cleaned whole-genome sequencing data of 37 samples, and the results were subsequently manually curated. Three published mitogenomes of *Rhynchium* were downloaded from NCBI and added to the mitogenome analyses (Appendix A). In total, 13 PCGs and 2 rRNAs from 40 samples were extracted by mitoz-tools (a program package within MitoZ v3.4) [25]. Multiple sequence alignments of the 13 PCGs were conducted using MAFFT v7.505 [26] within PhyloSuite v1.2.3 [27,28] with the L-INS-i strategy. For the alignments of the two rRNAs, the G-INS-i strategy in MAFFT was employed. In addition, PCGs were refined using MACSE v2.06 [29] for multiple sequence alignment in PhyloSuite. Ambiguously aligned fragments of the 13 PCGs and 2 rRNAs were removed using Gblocks 0.91b [30]. Concatenation of gene sequences was achieved using the ’Concatenate Sequence’ function in PhyloSuite, which was then used for the subsequent construction of the phylogenetic tree.

Additionally, the *COI* sequences were extracted for the subsequent construction of a phylogenetic tree, and the genetic distances of *COI* sequences of the three species were calculated based on the K2P (Kimura-2-parameter) [31] model using MEGA11 [32]. Species delimitation analyses based on *COI* sequences were conducted using ABGD (Automated Barcode Gap Discovery) [33].

### 2.4. Genome Assembly and Single-Copy Orthologous Dataset Generation

Genome assembly and the identification of single-copy orthologs (USCOs) were carried out using the cleaned whole-genome sequencing data from 37 samples and two outgroups, *Allorhynchium chinense* (accession number: SAMN36845336) and *A. argentatum* (SAMN36845335). Firstly, BBTools v38.96 [34] was employed for quality control and normalization. Subsequently, SPAdes v3.15.5 [35] was used for genome assembly. Finally, GapCloser v1.12 [36] was utilized to fill in the gaps. The assessment of genomic completeness was performed using BUSCO v5.4.3 [22] against the Hymenoptera database.

Universal single-copy orthologs (USCOs) were extracted from genomes using BUSCO v5.4.3 against reference Hymenoptera gene sets (n = 5991). The USCO amino acid and nucleotide sequences were then utilized for subsequent analyses. The USCO amino acid and nucleotide sequences of each locus were aligned using the L-INS-I strategy in MAFFT v7.490. Subsequently, they were trimmed using trimal v.1.4.1 [37] to eliminate gaps and ambiguous sites. The trimmed alignments were concatenated by FASConCAT-g v1.05 [38], generating matrix USCO90 with 90% completeness.

### 2.5. Phylogenetic Analyses

Phylogenetic analyses were performed based on four datasets: COI sequences, 13 PCGs and 2 rRNAs in the mitochondrial genomes, USCO nucleotide matrix of 90% completeness (USCO90_fna), and USCO amino acid matrix of 90% completeness (USCO90_faa). According to our previous research [15,16], the genera *Allorhynchium* and *Rhynchium* are closely related in terms of evolutionary relationship. Therefore, *Allorhynchium chinense* and *A*. *argentatum* were selected as the outgroups. Phylogenetic analysis was conducted using IQ-TREE [39]. The most appropriate substitution model was chosen by ModelFinder [40]. The support for the resulting maximum likelihood (ML) tree was evaluated using UFBoot [41] and SH-aLRT [42], with 1000 replicates for each. For the first two datasets, ML analysis was inferred using IQ-TREE v2.2.0 in PhyloSuite. For the last two, partitioned [43] ML analysis was performed, with the full IQ-TREE v2.2.0.8 command: ‘iqtree2 -s example.phy -p example.nex -m MFP+MERGE --symtest-remove-bad -B 1000 -alrt 1000 --prefix -rcluster 10’.

## 3. Results and Discussion

### 3.1. Genome Assembly and Matrix Generation

In this study, a total of 37 genomes of the genus *Rhynchium* were assembled. However, the BUSCO integrity of two samples, namely GS_2 and SXLF2 (Figure 2), was below 50%, and they were removed for the subsequent phylogenetic analysis. Among the remaining 35 samples of the genus *Rhynchium*, the sequencing depth was between 26.44X and 78.85X. The assembled genome sizes spanned from 184.76 Mb to 232.39 Mb. The number of scaffolds ranged between 24,500 to 306,444, with the N50 from 4.36 kb to 134.32 kb. The max length fell within the interval of 129.66 kb to 1281.60 kb, and the GC content ranged from 36.87% to 37.36%. The USCO completeness ranged from 64.00% to 94.60% (corresponding to 3834 to 5668 loci). Subsequently, after sequence alignment, filter and trimming, a USCO90 matrix was created across all markers, and the lowest proportion of species representation reached 90%. It comprised 4202 USCOs and 2,007,299 amino acid sites, forming a comprehensive data set that holds great potential for further in-depth analyses in relevant research fields [35]. The fundamental statistical data regarding the genome assembly of the genus *Rhynchium* can be found in Appendix A.

### 3.2. Genetic Distance and Species Delimitation

The 40 *COI* sequences of the genus *Rhynchium* were imported into the ABGD online website. Under the Jukes–Cantor (JC69) model, the genus *Rhynchium* was delimited, yielding the initial and recursive partition shown in Figure 3. Among them, the initial partition relatively stably divided the 40 samples into three or four groups, while the recursive partition divided the 40 samples into five groups with over-partitioning. Therefore, the relatively stable initial partition was selected as the reference. When the prior value (*p* value) was between 0.0129 and 0.0599, the samples were divided into three different species: *Rhynchium* A, *Rhynchium* B, *Rhynchium* C; when the prior value (*p* value) was between 0.0010 and 0.0077, the samples were divided into four different species: *Rhynchium* A, *Rhynchium* B, *Rhynchium* C (1,2). The setting of the *p* value is related to the complexity and diversity of the data. If the numerous differences among the *COI* sequences, to some extent, represent differentiation between species, a smaller *p* value is appropriate as it can accurately capture these differences to delimit species. Conversely, if there is a lot of noise in the data or small differences among individuals (such as intraspecific variation), a larger *p* value can avoid over-delimitation and group these individuals with small differences into the same species [44]. Since the interspecific and intraspecific differences in *COI* in the genus *Rhynchium* were not significant, it was more reasonable to divide *Rhynchium* into three species using a larger *p* value (ranging from 0.0129 to 0.0599).

Based on the *COI* sequences, the genetic distances among the groups were calculated, as shown in Table 1 and Appendix A. The interspecific genetic distances were 13.99% between *Rhynchium* A and *Rhynchium* B, 12.77% between *Rhynchium* A and *Rhynchium* C, 7.41% between *Rhynchium* B and *Rhynchium* C, and 1.23% between *Rhynchium* C (1) and *Rhynchium* C (2). According to Hebert’s perspective, the average difference in *COI* sequences within the species is typically less than 2% [45], and our results suggest that *Rhynchium* C (1) and *Rhynchium* C (2) are the same species, *Rhynchium* C; *Rhynchium* A, *Rhynchium* B and *Rhynchium* C should be three species.

### 3.3. Phylogenomic Analyses

Four phylogenetic trees were reconstructed from the *COI* sequences, with 13 PCGs and 2 rRNAs in the mitochondrial genome, a USCO nucleotide matrix of 90% completeness (USCO90_fna), and a USCO amino acid matrix of 90% completeness (USCO90_faa). In all the trees, *Rhynchium* A, B and C were clustered into monophyletic clades with robust support, forming an independent evolutionary lineage, although the relationships within a few respective populations were inconsistent. *Rhynchium* A diverged first and was sister to the clade of sister species *Rhynchium* B and *Rhynchium* C (Figure 4). In addition, the results support that the vast majority of our specimens belong to *Rhynchium* C, which was divided into two clades.

### 3.4. Morphological Analysis

Commonly used distinguishing features, including inner side of posterior ocellus bulge (Figure 5E–H and Figure 6A–I), width of cephalic foveae in female (Figure 5E–H), punctations of both mesoscutum and scutellum (Figure 5I–L), and depression of metanotum (Figure 5I–L), were selected for identification in the subfamily Eumeninae [14]. Based on the aforementioned analyses of genetic distance and molecular phylogenesis, *Rhynchium* A, B and C should be valid species, verifying that the above morphological features we selected are reliable for preliminary grouping in the genus *Rhynchium*. In addition, genitalia (Figure 6J–O) of male specimens were compared to find that those of *Rhynchium* B, with stouter aedeagus apically and bigger and rounder emargination of volsella submedianly (Figure 6J,K), are a little different from those (Figure 6L–O) of *Rhynchium* C. There are almost no differences within *Rhynchium* C between those transitional and confusing specimens. With a variety of color patterns in the genus *Rhynchium*, color was not used as an independent or unique morphological distinguishing feature and its variations will be shown in Geographical Distribution.

By examining our specimens (CNU) and those from India, Sri Lanka and Pakistan deposited in NHM, *Rhynchium* A, B and C were in order recognized as *R. carnaticum* (Fabricius, 1798), *R. quinquecinctum* (Fabricius, 1787) and *R. brunneum* (Fabricius, 1793), of which *R. carnaticum* is newly recorded in China. According to the key below, these three species can be distinguished from each other with morphological features rather than colors.

Key to the Chinese species of the genus *Rhynchium*

1. (i) Inner side of posterior ocellus with bulge (Figure 5F–H), and posterior ocellus just partially visible in frontal view (Figure 6B–I). Cephalic foveae area wider than or as wide as the distance between posterior ocelli in female (Figure 5F–H): see 2

(ii) Inner side of posterior ocellus without bulge (Figure 5E), and posterior ocellus completely visible in frontal view (Figure 6A). Cephalic foveae area less than the distance between posterior ocelli in female (Figure 5E) ⋯ *R. carnaticum*, newly recorded in China

2. (i) Cephalic foveae area wider than the distance between posterior ocelli in female (Figure 5F). Mesoscutum posteriorly and scutellum dull, densely punctate (Figure 5J). Metanotum hardly compressed medially (Figure 5J) ⋯⋯⋯⋯⋯ *R. quinquecinctum*

(ii) Cephalic foveae area as wide as the distance between posterior ocelli in female (Figure 5G,H). Mesoscutum posteriorly and scutellum anteriorly impunctate, more or less polished (Figure 5K,L). Metanotum compressed medially (Figure 5K,L) ⋯⋯ *R*. *brunneum*

### 3.5. Geographic Distribution

According to the Chinese locations (Figure 7) of the more than 600 specimens, *R. carnaticum* is just confined to Hainan, and *R. quinquecinctum* is limited to the southern part of Yunnan rather than widely present in China [7]. In striking contrast, *R. brunneum* is widely distributed across China. This result overturns the existing distribution records of the genus *Rhynchium* in China [7,12]. Meanwhile, the color pattern of *R. brunneum* shows significant variations: some are distinctly mostly reddish-brown (Figure 5C,G,K), while others are yellowish-brown with significantly smaller spots (Figure 5D,H,L). Interestingly, the specimens mainly reddish-brown in color and with sparser punctations on the scutellum were from the southern Chinese provinces and were mostly clustered together into one clade in the phylogenetic trees (as indicated by red sample IDs in Figure 4), whereas the specimens with yellowish-brown markings and denser punctations on the scutellum were from the northern and central provinces and in another clade in the phylogenetic trees. In addition, a few specimens from the central region (as indicated by the samples with blue IDs in Figure 4) displayed a transitional state.

In the color patterns of bumblebees, Williams (2007) [46] found that their color patterns underwent a gradual darkening process from mid-latitudes towards high latitudes and dark hues tended to be more prevalent in tropical zones, high-altitude terrains, and high-latitude areas. In Williams’ research, the bumblebees with the darkest coloration were predominantly linked to tropical regions, whereas those with the lightest coloration were associated with moderately northern latitudes, where a cryptic function in arid grasslands is speculated. In our study, the specimens of *R. brunneum* from southern tropical regions exhibited a distinct red coloration, displaying an overall reddish-brown color. In contrast, the specimens of *R. brunneum* from the central and northern regions exhibited lighter overall markings with a yellowish-brown color. Some specimens more or less with both markings are distributed in the transitional zone from the South to the North of China. Therefore, we hypothesize that the color patterns of *R. brunneum* are as closely related to geographical location and living environment as those of bumblebees. The specific influencing factors of color patterns are not clear and further investigations will be helpful to understand the mechanism of markings in wasps.

## 4. Conclusions

This integrative taxonomic study newly disentangles the confusion about the Chinese species of the genus *Rhynchium*. With genetic distance, phylogenetic relationships, morphological comparison and geographical distribution, three species (*R. brunneum*, *R. carnaticum* and *R. quinquecinctum*) were delineated in China, with *R. carnaticum* being newly recorded; the boundaries of these sister species could be defined by morphological characteristics. In the present research, we initially described the geographical distributions of different color patterns of the widely distributed species *R. brunneum* and hypothesized that the color patterns of *R. brunneum* are as closely related to geographical location and living environment as those of bumblebees. As with the well-known rich color patterns in the family Vespidae, their underlying evolutionary mechanisms are unknown. Further exploring the color patterns of the species *R. brunneum* may be a key to reveal the evolutionary adaptation of this family.

## Figures and Tables

**Figure 1 insects-16-00217-f001:**
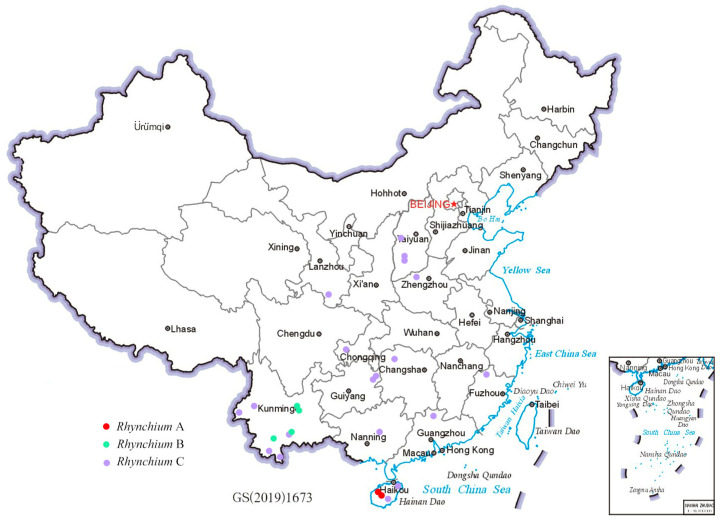
Geographical distribution of 37 sequenced samples in China.

**Figure 2 insects-16-00217-f002:**
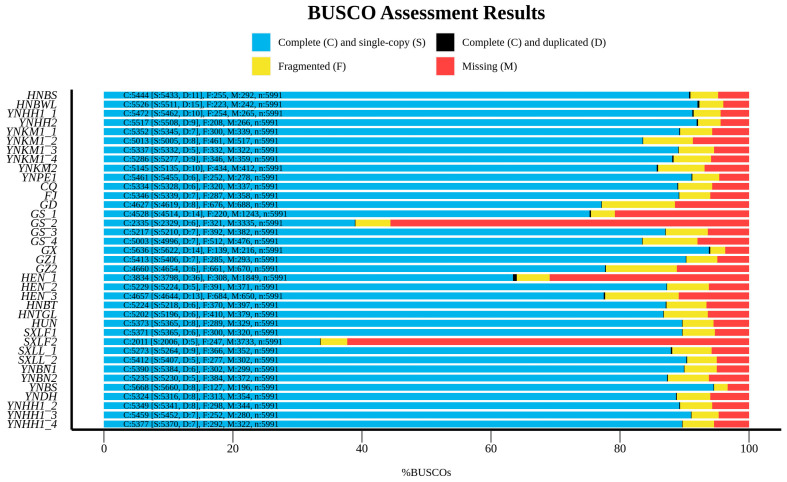
BUSCO completeness: complete (C) and single-copy (S, Light blue), complete (C) and duplicated (D, Black), fragmented (F, Yellow), and missing (M, Red).

**Figure 3 insects-16-00217-f003:**
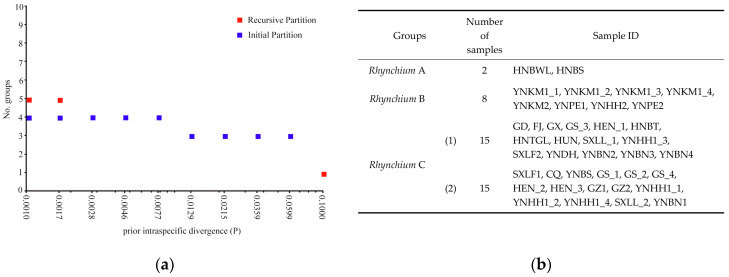
Automatic partition results from the Automatic Barcode Gap Discovery (ABGD) species delimitation of *Rhynchium* based on *COI* sequences: (**a**) Number of groups identified; (**b**) Detailed grouping results for *Rhynchium*.

**Figure 4 insects-16-00217-f004:**
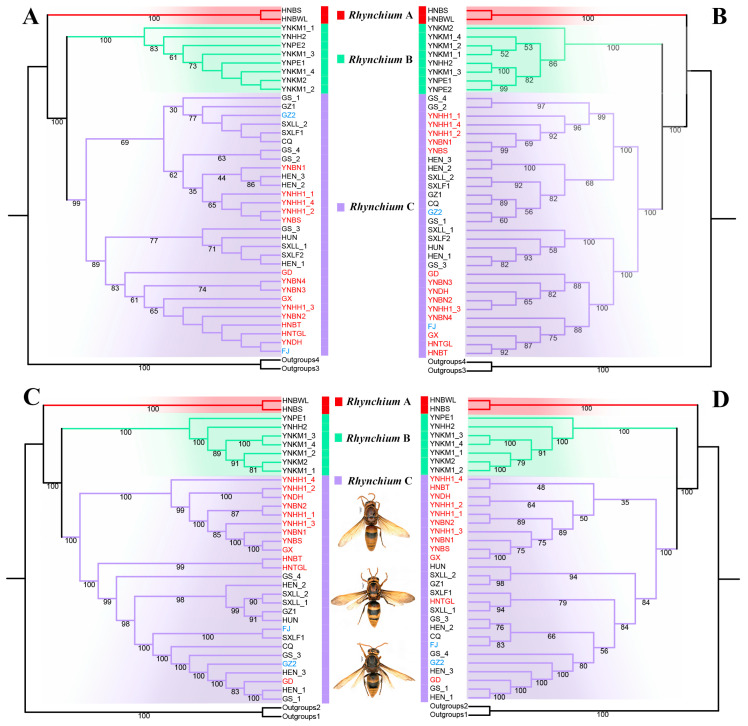
Phylogenomic analyses. (**A**) Phylogenetic tree constructed based on *COI*. (**B**) Phylogenetic tree constructed based on 13 PCGs and 2 rRNAs in the mitochondrial genome. (**C**) Phylogenetic tree constructed based on the USCO nucleotide matrix with 90% completeness (USCO90_fna). (**D**) Phylogenetic tree constructed based on the USCO amino acid matrix with 90% completeness (USCO90_faa). Branches of different colors represent different species: red for *Rhynchium* A, green for *Rhynchium* B, and purple for *Rhynchium* C. Among the *Rhynchium* C samples, those with IDs highlighted in red are from the south, while those with IDs highlighted in blue are from the central transitional zone.

**Figure 5 insects-16-00217-f005:**
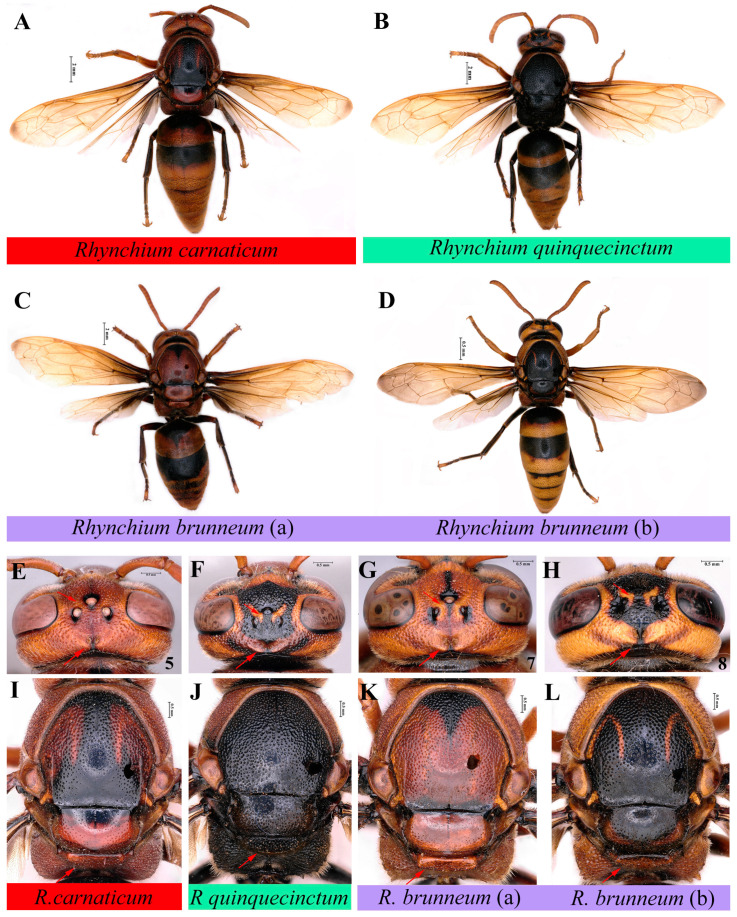
(**A**,**E**,**I**) *Rhynchium carnaticum*. (**A**) Habitus, dorsal view. (**E**) Head, dorsal view. (**I**) Mesosoma, dorsal view. (**B**,**F**,**J**) *Rhynchium quinquecinctum*. (**B**) Habitus, dorsal view. (**F**) Head, dorsal view. (**J**) Mesosoma, dorsal view. (**C**,**G**,**K**) *Rhynchium brunneum* (**a**). (**C**) Habitus, dorsal view. (**G**) Head, dorsal view. (**K**) Mesosoma, dorsal view. (**D**,**H**,**L**) *Rhynchium brunneum* (**b**). (**D**) Habitus, dorsal view. (**H**) Head, dorsal view. (**L**) Mesosoma, dorsal view.

**Figure 6 insects-16-00217-f006:**
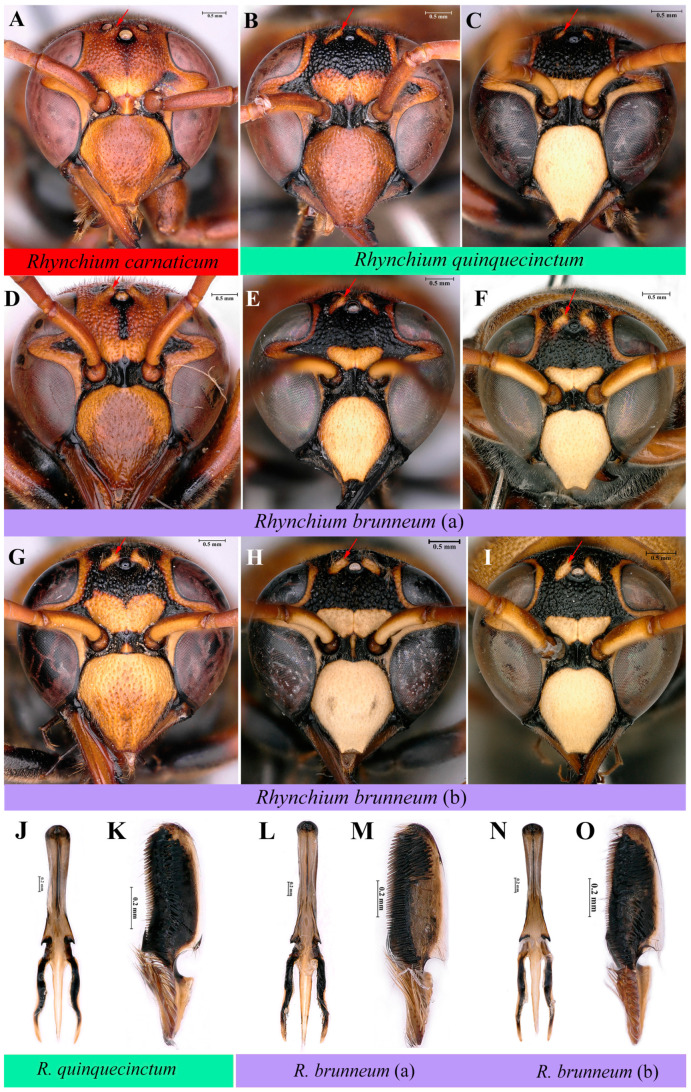
(**A**) *Rhynchium carnaticum*. Head, frontal view. (**B**,**C**,**J**,**K**) *Rhynchium quinquecinctum*. (**B**) Head, frontal view. (**C**) Head, frontal view. (**J**) Aedeagus, ventral view. (**K**) Volsella, ventral view. (**D**–**F**,**L**,**M**) *Rhynchium brunneum* (**a**). (**D**) Head, frontal view. (**E**,**F**) Head, frontal view. (**L**) Aedeagus, ventral view. (**M**) Volsella, ventral view. (**G**–**I**,**N**,**O**) *Rhynchium brunneum* (**b**). (**G**) Head, frontal view. (**H**,**I**) Head, frontal view. (**N**) Aedeagus, ventral view. (**O**) Volsella, ventral view.

**Figure 7 insects-16-00217-f007:**
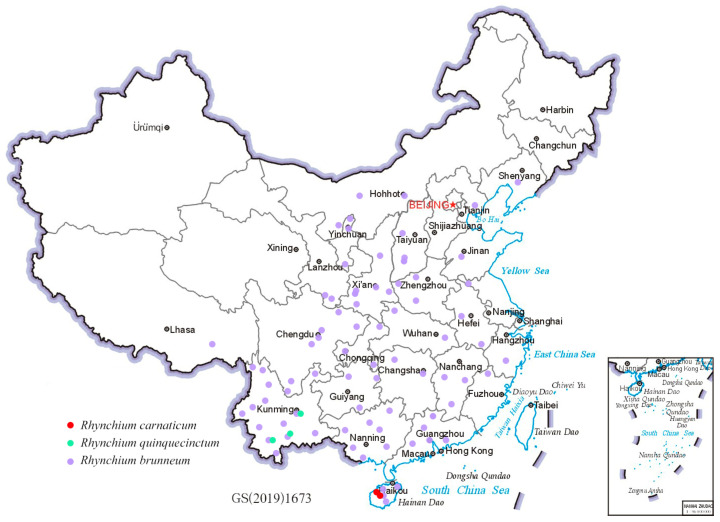
Species distribution map of the genus *Rhynchium* in China.

**Table 1 insects-16-00217-t001:** Genetic distance based on K2P and *COI* sequences.

Species	Number of Samples	Mean Distance Within Species	Mean Distance Between Species		
			*Rhynchium* A	*Rhynchium* B	*Rhynchium* C (1)
*Rhynchium* A	2	0	-	-	-
*Rhynchium* B	8	0.0003	0.1399	-	-
*Rhynchium* C (1,2)	30	0.0069	0.1277	0.0741	-
*Rhynchium* C (1)	15	0.0011	0.1267	0.0740	-
*Rhynchium* C (2)	15	0.0009	0.1287	0.0741	0.0123

## Data Availability

The data presented in this study are openly available from the National Center for Biotechnology Information at https://www.ncbi.nlm.nih.gov (accessed on 16 December 2023), accession numbers: PRJNA1207318 (https://www.ncbi.nlm.nih.gov/sra/PRJNA1207318 (accessed on 16 December 2023, activation date 7 January 2029).

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
