# Peer review of "An Integrative Phylogenetic Analysis of the Genus Rhynchium Spinola (Hymenoptera: Vespidae: Eumeninae) from China Based on Morphology, Genomic Data and Geographical Distribution"

_insects, 2025, doi:10.3390/insects16020217_

Round 1
Reviewer 1 Report
Comments and Suggestions for Authors
This manuscript presents a lot of work to show that there are three Chinese species of Rhynchium. In view of the fact that the three species are distinguishable by morphology, the reader might question why all the molecular work was done - it wasn't necessary. The manuscript would certainly be more concise without the molecular part. Be that as it may, specific suggestions for minor revisions follow:
line 55, "marked considerable" choose one word, not both.
line 59, remove the word "Whereas."
line 65, Selis is not the author of dantici, and it was not described in 2014.
line 98, "generic" is the wrong word; you are writing about specific differences.
line 160, Rhynchium is not an outgroup, it is the ingroup.
line 294, use punctations, not punctuations.
Comments on the Quality of English LanguageSee specific suggestions above.
Author Response
Comments 1: In view of the fact that the three species are distinguishable by morphology, the reader might question why all the molecular work was done - it wasn't necessary. The manuscript would certainly be more concise without the molecular part.
Response 1: Thanks for the reviewer’s comment. About the question why all the molecular work was done, we explained the reason in “Introduction” and the morpholgy used in our study was not used before in the genus. So, the molecular work was necessary to verify and distinguish the confusing species.
Comments 2: line 55, "marked considerable" choose one word, not both.
Response: Thanks for the reviewer's valuable comments. The words "marked" and "considerable" overlap, both emphasizing considerable degree. To avoid redundancy, we've chosen "marked" in the revised MS.
Comments 3: line 59, remove the word "Whereas."
Response 3: According to the reviewer's comment, we have removed the word "Whereas".
Comments 4: line 65, Selis is not the author of dantici, and it was not described in 2014.
Response 4: Thank you for your correction. we have corrected as Euodynerus dantici (Rossi, 1790) in the revised MS.
Comments 5: line 98, "generic" is the wrong word; you are writing about specific differences.
Response 5: Thank you for your correction. "Generic" has been changed to "specific".
Comments 6: line 160, Rhynchium is not an outgroup, it is the ingroup.
Response 6: Indeed, we mistakenly wrote "Rhynchium" as the outgroup. We have corrected it by classifying "Rhynchium" as the ingroup, and the outgroups we have chosen are Allorhynchium chinense and A. argentatum. Corresponding adjustments have been made in the text.
Comments 7: line 294, use punctations, not punctuations.
Response 7: Thank you for your correction. It has been changed to "punctuation". Thank you for your review.
Reviewer 2 Report
Comments and Suggestions for Authors
Wasps of the subfamily Eumeninae are poorly studied genetically and this paper is a valuable piece of work. At least one reference can be added and there are several points that should be rewritten. Please see the attached file for detail.
(Please take into account that I am a wasp specialist but not an expert in the bioinformatics, and also not a native English speaker).

Author Response
Comments 1: References about pest control?
Response 1: According to your suggestion, we have added two references on pest control.
- Klein, A.M.; Steffan-Dewenter, I.; Tscharntke, T.Foraging trip duration and density of megachilid bees, eumenid wasps and pompilid wasps in tropical agroforestry systems. Anim. Ecol. 2004, 73, 517–525. https://doi.org/10.1111/j.0021-8790.2004.00826.x.
- Dang, H.T.; Nguyen, L.T.P. Nesting biology of the potter wasp Rhynchium brunneum brunneum (Fabricius, 1793) (Hymenoptera: Vespidae: Eumeninae) in North Vietnam. Asia-Pac. Entomol. 2019, 22, 427-436. https://doi.org/10.1016/j.aspen.2019.02.003.
Comments 2: (Rossi, 1790) not Selis, 2024!
Response 2: Thank you for your correction. It has been updated to Euodynerus dantici (Rossi, 1790).
Comments 3: The genus Rhynchiumcannot be an outgroup, it in the ingroup. And please indicate species used as an outgropt, not only genera.
Response 3: Indeed, we mistakenly wrote "Rhynchium" as the outgroup. We have corrected as “the genera Allorhynchium and Rhynchium are closely related in terms of evolutionary relationship. So, Allorhynchium chinense and A. argentatum were selected as the outgroups”.
Comments 4: This key looks like based on females only. This should be indicated, or male characters should be included.
Response 4: Thank you for your comments. In the key, only “cephalic foveae area” is based on female and other characters are based on both female and male. We have indicated in the revised MS.
Comments 5: This is very questinable. Northern specimens are darker than southern becasue they are more black, whith less extensive brigth pattern. How darker markings can help under more active sun? Well, they (markings) are darker but the general coloration is not darker but brighter. And in bumblebees, there is a color pattern made by differently colored setation, not the cuticula surface as in the wasps. Thus, they cannot be compared so directly. Please rewrite this.
Response 5: Thank you for your valuable comment. According to your suggestions, we have rewritten the relevant content as below. "In our study, the specimens of R. brunneum from southern tropical regions exhibit a distinct red coloration, displaying an overall reddish-brown color. In contrast, the specimens from the central and northern regions of R. brunneum exhibited lighter overall markings with yellowish-brown color. Some specimens more or less with both markings are distributed in the transitional zone from South to North of China. Therefore, we hypothesize that color patterns of R. brunneum are as closely related to geographical location and living environment as that of bumblebees. The specific influencing factors of color patterns are not clear and further investigations will be helpful to understand the mechanism of markings in wasps."
Reviewer 3 Report
Comments and Suggestions for Authors
This is a good and informative work in the taxonomic field, which shows the advantages of the integrative method, with a decisive role for molecular data in dispelling doubts about interspecific limits. Only the final part, entitled "Geographic analysis", seems out of place, adds little and, in fact, no analysis is done, dealing only with the description of the geographic distribution of color patterns. The analogy made to Williams' work with bees seems inadequate, there being nothing in particular in the data that justifies the supposition that the patterns are conditioned by climatic variables (which the authors do not even know what these would be, limiting themselves to saying that "further investigations will be helpful to understand ...")
Comments on the Quality of English LanguageThe English is reasonable for the most part, but additional proofreading would be beneficial to clarify some problems with word choice and sentence articulation
Author Response
Comments 1: Only the final part, entitled "Geographic analysis", seems out of place, adds little and, in fact, no analysis is done, dealing only with the description of the geographic distribution of color patterns. The analogy made to Williams' work with bees seems inadequate, there being nothing in particular in the data that justifies the supposition that the patterns are conditioned by climatic variables.
Response 1:
As the reviewer has said, it is not possible to identify the environmental factors that affect the different color pattern in our current study. So, we have changed the title "Geographic analysis" to “Geographic distributions”. We will explore this question further in our follow-up research.
About "the analogy made to Williams' work with bees", we have rewritten as below.
"In our study, the specimens of R. brunneum from southern tropical regions exhibit a distinct red coloration, displaying an overall reddish-brown color. In contrast, the specimens from the central and northern regions of R. brunneum exhibited lighter overall markings with yellowish-brown color. Some specimens more or less with both markings are distributed in the transitional zone from South to North of China. Therefore, we hypothesize that color patterns of R. brunneum are as closely related to geographical location and living environment as that of bumblebees. The specific influencing factors of color patterns are not clear and further investigations will be helpful to understand the mechanism of markings in wasps."